# Research of Global Tilt and Functional Independence: Insights into Spinal Health of Older Women

**DOI:** 10.3390/bioengineering11050493

**Published:** 2024-05-16

**Authors:** Yu-Chieh Chiu, Ping-Chiao Tsai, Ssu-Hsien Lee, Wen-Tien Wu, Tzai-Chiu Yu, Ru-Ping Lee, Ing-Ho Chen, Jen-Hung Wang, Kuang-Ting Yeh

**Affiliations:** 1School of Medicine, Tzu Chi University, Hualien 970, Taiwan; 107311106@gms.tcu.edu.tw (Y.-C.C.); 107311154@gms.tcu.edu.tw (P.-C.T.); 107311120@gms.tcu.edu.tw (S.-H.L.); timwu@tzuchi.com.tw (W.-T.W.); feyu@tzuchi.com.tw (T.-C.Y.); ihchen@tzuchi.com.tw (I.-H.C.); 2Department of Orthopedics, Hualien Tzu Chi Hospital, Buddhist Tzu Chi Medical Foundation, Hualien 970, Taiwan; 3Institute of Medical Sciences, Tzu Chi University, Hualien 970, Taiwan; fish@gms.tcu.edu.tw; 4Department of Medical Research, Hualien Tzu Chi Hospital, Buddhist Tzu Chi Medical Foundation, Hualien 970, Taiwan; paulwang@tzuchi.com.tw; 5Department of Medical Education, Hualien Tzu Chi Hospital, Buddhist Tzu Chi Medical Foundation, Hualien 970, Taiwan; 6Graduate Institute of Clinical Pharmacy, Tzu Chi University, Hualien 970, Taiwan

**Keywords:** spinal alignment, bone mineral density, muscle strength, global tilt, elderly women, sagittal balance, functional independence

## Abstract

Spinal alignment intricately influences functional independence, particularly in older women with osteopenia experiencing mild neck and back pain. This study elucidates the interplay between spinal alignment, bone mineral density (BMD), and muscle strength in elderly women presenting with mild neck and back pain. Focusing on a cohort of 189 older women, we examined the associations among global tilt (GT), coronal and sagittal alignment, BMD, grip strength, and functional independence as gauged by the Barthel index. Our findings indicate significant associations between functional capacity and grip strength, bone density, GT, and pelvic tilt (PT). Elderly women with a Barthel Index above 80 demonstrated higher grip strength and better bone quality, reflected by less negative average T scores. These individuals also exhibited lower values of GT and PT, suggesting a better sagittal alignment compared to those with a Barthel index of 80 or below. The results highlight that deviations in GT and PT are significantly associated with decreased functional independence. These insights emphasize the importance of maintaining optimal spinal alignment and muscle strength to support functional independence in elderly women. This study underscores the potential for targeted interventions that improve postural stability and manage pain effectively in this vulnerable population.

## 1. Introduction

Spinal alignment plays a key role in comprehensive management for patients with spinal disorders [1]. The importance of spinal alignment is further underscored by its profound effect on the well-being of individuals with adult spine deformities (ASDs) [2]. In patients with ASDs, compromised spinal alignment is associated with a reduction in quality of life [3]. Global tilt (GT) is a distinct and invaluable measure for evaluating global sagittal alignment [4]. Unlike other techniques, such as whole-spine radiography, GT is not affected by positional variations; thus, it has diagnostic reliability [4]. Notably, GT is used as both a diagnostic and a prognostic tool, particularly in post-ASD surgical outcomes. Therefore, GT plays a pivotal role in the therapeutic landscape [5]. With respect to patient-reported outcomes, such as the Oswestry Disability Index (ODI) score, reduced GT exhibits a dynamic interplay with other global sagittal alignment parameters, such as the T1 pelvic angle and sagittal vertical axis (SVA). These parameters vary with age and gender; that is, they tend to progressively increase with aging and tend to be higher in females than in males [6].

Research on GT has primarily focused on investigating the complex relationship between thoracolumbar spine dynamics and pelvic sagittal alignment [7]. However, certain aspects of this complex relationship have been insufficiently explored. For example, the compensatory mechanism underlying cervical lordosis plays a pivotal role in the overall dynamics of the spine [8], particularly among aging individuals. This multifaceted relationship between cervical lordosis, aging, and spinopelvic complex equilibrium requires comprehensive evaluation; the present study was conducted to bridge this gap in the literature. Recognizing the multifarious determinants capable of influencing spinopelvic equilibrium is essential. Factors such as bone quality [9] and muscular strength [10] can contribute to the intricate framework of spinal health by potentially modulating this equilibrium. This intricate framework inspired the current analysis of the nuanced associations between cervical and lumbopelvic sagittal parameters.

This study focused on a specific demographic, namely older women with mild neck and back discomfort. Within this population, the interplay between GT, body mass index (BMI), and osteoporosis plays a central role in spinal health that requires extensive exploration. In this study, we analyzed the complex interplay between these factors and their collective effect on spinopelvic equilibrium, providing valuable insights into the spinal health challenges experienced by this population.

## 2. Materials and Methods

This cross-sectional study was approved by the Institutional Review Board of our hospital. After providing written informed consent, women aged 65 or older who were seeking consultations for bone health between August 2019 and July 2023 were recruited for bone health evaluations from the orthopedic department of our hospital. These patients underwent dual-energy X-ray absorptiometry for bone mineral density (BMD) measurements. The inclusion criteria were as follows: 1. female gender older than 65 years; 2. having the intermittent minimal-mild neck or back pain [11], with a pain severity measured between 0 and 3 on the Visual Analog Scale (VAS); 3. having a normal walking status [12]; and 4. having low bone mass, defined by a T score of below −1.0 at specific locations, including the lumbar spine (average), right hip (femoral neck or total femur), and left hip (femoral neck or total femur). The exclusion criteria were as follows: 1. those who previously underwent spine surgery, femoral neck fracture, artificial hip or knee replacement, or experienced acute spinal trauma within the previous 6 months; 2. those who had a history of stroke or had a history of a neurological disorder such as Parkinson’s or Alzheimer’s disease; and 3. those who cannot tolerate or agree to receiving the whole-spine standing lateral view X-ray examination [13].

Upon agreeing to participate in the study, the patients provided information on their age, menopausal status, and BMI. We then arranged full-length standing anteroposterior and lateral radiography examinations for them for further measurement of the radiographic parameters [14], measured their grip strength, which was determined as the mean of triplicate measures of bilateral hand strength, and evaluated their Barthel index score, which is a standardized assessment tool that quantifies an individual’s functional independence by scoring their ability to perform essential activities of daily living [15].

Coronal malalignment was defined as a Cobb angle ≥ 30 degrees measured from long-standing anteroposterior plain radiographs [16]. Both global sagittal parameters (GT and SVA) and regional parameters (upper cervical lordosis (UCL), middle cervical lordosis (MCL), lower cervical lordosis (LCL), C7 slope, upper thoracic kyphosis (UTK), lower thoracic kyphosis (LTK), upper lumbar lordosis (ULL), lower lumbar lordosis (LLL), pelvic incidence (PI), sacral slope (SS), and pelvic tilt (PT)) were measured using long-standing lateral plain radiographs. The sagittal parameters were introduced and defined as follows [5,17,18,19] (Figure 1A–D): GT is determined by the angle between C7, the center of the sacrum, and the line between the center of the femoral heads and the center of the sacrum (Figure 1A). SVA is the length of a horizontal line connecting the posterior superior sacral end plate to a vertical plumbline dropped from the centroid of the C7 vertebral body (Figure 1A). The cervical lordotic angle was divided into UCL, MCL, and LCL. UCL is the angle, measured in degrees, formed between the cranial line (C0) and the lower endplate of C2 (Figure 1B). MCL is the angle, measured in degrees, formed between the lower endplate of C2 and the lower endplate of C5 (Figure 1B). LCL is the angle, measured in degrees, formed between the lower endplate of C5 and the lower endplate of C7 (Figure 1B). C7 slope is determined by measuring the angle between a horizontal reference line and a line that runs parallel to the upper endplate of C7 (Figure 1B). The thoracic kyphotic angle was divided into UTK and LTK. UTK is the angle, measured in degrees, formed between the lower endplate of T5 and the lower endplate of T9 (Figure 1C). LTK is the angle, measured in degrees, formed between the lower endplate of T9 and the lower endplate of T12 (Figure 1C). The lumbar lordotic angle was divided into ULL and LLL. ULL refers to the angle, measured in degrees, formed between the lower endplate of L1 and the lower endplate of L4 (Figure 1C). LLL is the angle, measured in degrees, formed between the lower endplate of L4 and the upper endplate of S1 (Figure 1C). Sacral slope is the angle between the upper endplate of S1 and the horizontal reference line (Figure 1D). PT is the angle between a vertical reference line and a line from the midpoint of the sacral endplate to the femoral rotational axis (Figure 1D). Four physicians independently measured the parameters from the plain films. The intraclass association coefficient of the measurements was 0.88 and 0.81 for intraobserver and interobserver agreement, respectively.

All statistical analyses were conducted using IBM SPSS Statistics for Windows (version 23.0; IBM, Armonk, NY, USA). Chi-square tests and independent *t* tests/Wilcoxon rank-sum tests (depending on whether the normality assumption holds or not) were used to evaluate the associations between demographic characteristics and the Bathel index score. Simple and multiple linear regression analyses were conducted to examine the associations between the Barthel index and grip strength, the average T score, and GT, along with other clinical parameters. The same analysis was conducted to investigate the associations between GT and both the average T score and grip strength, as well as other clinical and sagittal parameters. All *p* values were obtained using two-sided tests, with *p* < 0.05 indicating statistical significance. According to Bonett et al., the recommended sample size should be at least 50 + 8 × k. K represents the number of predictors considered in the multiple linear models. For the Barthel index, the minimum sample size should be 50 + 8 × 5 = 90. For global tilt angle, the minimum sample size should be 50 + 8 × 14 = 162. Thus, our sample size (*n* = 189) fulfilled the minimum requirement for sample size calculation [20].

## 3. Results

### 3.1. Demographics of Older Females

There were 189 elderly females with low bone mass, segmented by their Barthel Index scores into two groups: those with scores of 80 or below and those with scores above 80 (Table 1). This distinction helps in understanding the relationship between various clinical parameters and functional independence as measured by the Barthel Index. Of the total participants, 55 had a Barthel index of 80 or less, and 134 scored above 80. The overall average age was 69.41 years, with the lower Barthel group being slightly older (70.63 years) compared to the higher score group (68.90 years), although this age difference was not statistically significant (*p* = 0.140) (Table 1). Both groups had a similar duration of post-menopausal period, averaging around 19.58 years across the cohort, with no significant difference (*p* = 0.956). There was a slightly higher average BMI in the group with a higher Barthel score (24.48) compared to those with a lower score (23.82), but this difference was not statistically significant (*p* = 0.255). Grip strength was significantly higher in the group with Barthel scores over 80 (20.01) compared to those with scores of 80 or less (16.66), indicating an association between higher physical strength and better functional performance (*p* < 0.001) (Table 1). This measure of bone density was significantly higher (indicating better bone quality) in the higher Barthel score group (−1.67) compared to the lower score group (−2.10), with a *p*-value of 0.002. Both GT and SVA showed significant differences between the groups (Table 1). Higher GT (24.80) and SVA (45.99) were observed in participants with lower Barthel scores, indicating more pronounced spinal misalignment associated with decreased functional independence (*p* < 0.001 for GT and *p* = 0.003 for SVA). Similarly, PT was significantly higher in the lower Barthel group (21.92), further illustrating the relationship between pelvic alignment and functional capacity (*p* < 0.001) (Table 1).

### 3.2. Factors Associated with the Barthel Index

The Barthel index is utilized as a measure of functional independence, assessing the ability of individuals to perform daily activities. Table 2 lists both crude and adjusted beta coefficients (β) with their 95% confidence intervals (CI) and corresponding *p*-values, which elucidate the strength and statistical significance of these relationships under a multiple linear regression model statistical analysis. The postmenopausal period initially appeared to have a minimal and non-significant negative association with the Barthel index in the crude analysis (β = −0.09, CI: −0.21 to 0.03, *p* = 0.140), suggesting a negligible effect of the duration of the postmenopausal period on functional independence, which remained consistent after adjustment. BMI showed a non-significant positive association with the Barthel index in the crude analysis (β = 0.17, CI: −0.09 to 0.44, *p* = 0.194). This indicates that while there might be a slight tendency for higher BMI to be associated with better functional scores, the relationship does not reach statistical significance, potentially due to variability within the cohort or other confounding factors. Grip strength displayed a significant positive relationship with the Barthel index, indicating that greater muscle strength is associated with better functional independence. This relationship was strong and statistically significant both before (β = 0.68, CI: 0.50 to 0.86, *p* < 0.001) and after adjustments (β = 0.53, CI: 0.35 to 0.71, *p* < 0.001). Reflecting bone density, the average T score also showed a significant positive association with the Barthel index. Individuals with higher T scores, indicating better bone quality, tended to have higher functional scores. This relationship was robust in both the crude (β = 2.58, CI: 1.57 to 3.60, *p* < 0.001) and adjusted analyses (β = 1.49, CI: 0.56 to 2.43, *p* = 0.002). Interestingly, a higher global tilt angle, indicating worse spinal alignment, was associated with lower functional scores. This negative association was significant in both crude (β = −0.25, CI: −0.35 to −0.16, *p* < 0.001) and adjusted models (β = −0.16, CI: −0.25 to −0.08, *p* < 0.001), underscoring the impact of spinal alignment on functional capacity. The adjusted R-squared value of 0.33 indicates that approximately 33% of the variance in Barthel index scores among the participants can be explained by these factors. This substantial level of explained variance highlights the critical roles that physical health metrics such as muscle strength, bone density, and spinal alignment play in the functional independence of elderly women.

### 3.3. Factors Associated with GT

The results of a multiple linear regression model statistical analysis of clinical and radiographic factors associated with the GT have been presented in Table 3. The table reports both crude and adjusted β with 95% CI and *p*-values, elucidating the strength and significance of associations between GT and multiple clinical or radiographic factors after controlling for potential confounders. Initially, the menopause period showed a moderate positive association with GT in the crude analysis (β = 0.30, CI: 0.14–0.46, *p* < 0.001), suggesting that as the duration post-menopause increases, so does the GT. However, this association diminished and became statistically insignificant after adjustments (β = 0.07, CI: −0.07–0.21, *p* = 0.312), indicating that other variables in the model may mitigate this effect. BMI showed no significant association with GT in both crude (β = 0.05, CI: −0.32–0.43, *p* = 0.780) and adjusted analyses, suggesting that BMI alone does not significantly affect spinal curvature in this group. Grip strength exhibited a negative relationship with GT, where higher grip strength was associated with a decrease in GT, although this relationship was significant only in the crude analysis (β = −0.47, CI: −0.75–−0.18, *p* = 0.002) and not after adjustment (β = −0.22, CI: −0.46–0.02, *p* = 0.076). The presence of coronal malalignment showed a strong and significant association with increased GT, with both crude (β = 5.35, CI: 2.40–8.31, *p* < 0.001) and adjusted values (β = 2.33, CI: 0.02–4.79, *p* = 0.048) indicating a substantial impact on global tilt. A lower T score, indicating poorer bone quality, was initially found to be associated with an increased GT (β = −2.29, CI: −3.79–0.79, *p* = 0.003); however, this association was not significant after adjustment (β = −0.82, CI: −2.05–0.41, *p* = 0.190). Various sagittal parameters, such as SVA, C7 slope, and LLL, also demonstrated significant relationships with GT. Particularly, SVA and LLL were significantly associated with GT in both crude and adjusted models, indicating their influence on spinal balance and alignment. The analysis also shows an adjusted R-squared value of 0.47, suggesting that about 47% of the variability in GT among participants can be explained by the factors included in the model. This high level of explained variance underscores the importance of the factors studied in determining spinal alignment and emphasizes the multifactorial nature of spinal health in the elderly.

## 4. Discussion

To our knowledge, this is the first study to investigate the associations between spinopelvic parameters and BMI, BMD, and grip strength. To address the gap in the literature, we focused on the major patient group in this aged society, mild-symptomatic older females. Our findings revealed that the Barthel index was associated with BMD, GT, and grip strength and that GT was associated with coronal malalignment, SVA, and LLL within this cohort. Overall, these findings provide valuable insights into improving the function of and offering advice for this particular population.

The findings of Table 1 in our study reveal significant associations between functional independence, as assessed by the Barthel index, and various clinical parameters in elderly women with low bone mass. Notably, grip strength, global tilt (GT), and pelvic tilt (PT) appear closely linked to functional outcomes. Higher grip strength is associated with better functional independence (*p* < 0.001), echoing research by Bohannon (2019), who highlights grip strength as a predictor of health and survival in older populations, underscoring its importance as a simple yet effective prognostic indicator for clinical assessments [21]. Furthermore, our results indicate that worse spinal alignment, represented by higher GT and SVA values, is associated with lower functional scores (*p* < 0.001 for GT and *p* = 0.003 for SVA). This aligns with the findings of Glassman et al. (2005), who report that sagittal balance is crucial for maintaining functional abilities in the elderly, as deviations can significantly impact the quality of life by increasing the risk of falls and reducing mobility [22]. Similarly, the significant difference in PT between the two Barthel index groups (*p* < 0.001) supports findings by Lafage et al. (2019), suggesting that pelvic tilt adjustments are compensatory mechanisms in response to spinal deformities, which can influence postural stability and functional capacity [23]. These insights emphasize the importance of comprehensive spinal evaluations in the management and therapeutic targeting of elderly patients at risk of functional decline due to musculoskeletal disorders. We also observed a significant association between GT and coronal malalignment. This finding is consistent with those of previous studies, which have reported an association between GT and coronal malalignment [1,24], emphasizing the interconnectedness of sagittal and coronal imbalances. Bao et al. reported an association between lower-end vertebral disc degeneration and sagittal imbalance in patients with degenerative lumbar scoliosis [25]. These findings indicate that disc degeneration may be a contributor to sagittal imbalance and thus contribute to the association between sagittal alignment and coronal malalignment.

Similar to the ODI in similar investigations [26], the Barthel index was key in evaluating patient function in the current study. A significant association was observed between the Barthel index and GT, average T score, and grip strength. Since our GT results are consistent with those of previous studies [6,27], low BMD and grip strength appeared to play important roles between GT and the functional scores among this mild-symptomatic patient group. Because certain concerns may arise regarding BMD in older women, we evaluated the lumbar and hip T scores of our patients to analyze the association between the average T score and the Barthel index. This association has been previously investigated in patients undergoing arthroplasty [28] and patients with sarcopenia [29]. According to González Silva et al., a major association exists between functional outcomes and fragility fracture risk, including BMD [30]. These findings from the literature and our study revealed the importance of taking BMD and muscle strength into consideration when studying the sagittal alignment of older adults. In this study, grip strength, which represents concurrent overall strength [21,31,32,33], was used as an indicator of skeletal muscle strength. In a previous study, Inoue et al. confirmed the predictive value of evaluating grip strength upon hospital admission for patients with hip fractures [34]. Briggs et al. reported that, compared with women without low back pain, those with chronic low back pain had significantly lower back muscle endurance [35]. Overall, these findings indicate that muscle strength is a valuable indicator of functional performance in both clinical assessments and predictive models.

Examinations of sagittal alignment and spinal deformity, which are novel spinopelvic parameters [4,6], indicate that GT may be beneficial, which may indicate the existence of associations that warrant further exploration. In this study, we discovered that GT is associated with coronal malalignment, SVA, and low LLL. Notably, GT is relatively stable against changes in patient positioning, accommodating variations in both spinal and pelvic alignment [4,6,25]. Our findings indicate that GT is not associated with grip strength or BMD, confirming its value as a metric that remains unaffected by variations in patient muscle strength and bone density. In a previous cohort study, Banno et al. argued that GT is associated with both age and gender [6]. However, in an extensive cross-sectional study, Charles et al. contradicted these findings and suggested that GT does not significantly associate with gender [19]. In the present study, we examined patients without coronal malalignment and discovered that our GT values were close to those reported by Charles et al. and Banno et al. for the same age group, which confirms that our results are consistent with those of previous studies [6,25].

Overall, our findings highlight multiple associations between the Barthel index and GT, BMD, and grip strength and between GT and coronal malalignment, SVA, and low LLL. These findings indicate that improving patient function involves not only addressing low bone mass but also promoting engagement in exercise or rehabilitation to maintain muscle strength and encouraging patients to assume a correct sitting posture to prevent coronal and sagittal malalignment. Braces are already widely used for patients with osteoporotic vertebral fractures [36]. In elderly individuals suffering from chronic back pain, the advent of novel brace designs might also offer a non-invasive strategy option prior to surgical interventions. Photogrammetric scanning systems have been demonstrated to be effective for the fabrication of custom-made spinal orthoses [37]. Furthermore, a sagittal realignment brace could offer substantial relief for individuals suffering from chronic low back pain [38]. Artificial intelligence systems determine the optimal approach to patient care by analyzing data and adjusting based on this information. Consequently, the data we compile may contribute to the vast pool of knowledge used in future deep learning and big data analytics. In cases that require surgical intervention, attention must be given to LLL correction. GT can be used to compare different patient subgroups and can be used as both a global alignment and proportion score [39] and a reference for surgical planning [40] for predicting mechanical complications after adult spinal deformity surgery.

While our study provides valuable insights into the associations between spinopelvic parameters and functional indices in older women with mild symptoms, it is important to acknowledge its limitations for a comprehensive understanding. First, methodological bias exists in this study. Although our study’s inclusion criteria focused on a specific age, gender, and demographic group, this approach was strategic. By targeting precisely defined subgroups, we aimed to mitigate the effects of variations in these characteristics, thereby enhancing the robustness and relevance of our findings to the target demographic. However, this specificity limits the generalizability of our results to broader populations. Although we employed standardized tools such as the Barthel index and DEXA scans for assessing functional independence and bone density, respectively, there might still be inconsistencies in how these measurements were applied or interpreted across different settings or examiners. While we adjusted for several potential confounders in our analyses, there may be additional unmeasured variables that could influence the outcomes, such as socioeconomic status, dietary factors, or previous medical interventions. Second, our study concentrated on the role of bone and muscle parameters in achieving spinal equilibrium in an upright posture. This approach inherently limits our exploration to the structural and functional aspects of bones and muscles, potentially overlooking other influential factors. Additionally, the impact of muscle fatigue on spinal alignment metrics was not considered. Muscle fatigue introduces a temporal dynamic that could affect the results, necessitating further research into dynamic spinal balance. Third, the omission of knee joint parameters, which undergo dynamic flexion and extension, is a notable limitation. These parameters have a significant connection to sagittal spinal alignment and, consequently, to patient outcomes. This exclusion restricts our understanding of the full spectrum of factors influencing spinal balance. Fourth, the study did not incorporate dynamic spinal balance parameters, thereby limiting our insights into the complexities of spinal balance dynamics. Future studies might benefit from using tools like the Dubousset Functional Test to assess functional ability and balance, providing a more comprehensive understanding of these dynamics [41]. Despite these limitations, our study still revealed a significant association between the functional parameter, the Barthel index, and GT, BMD, and grip strength. This underlines the multifaceted nature of spinal health, where structural alignment, bone quality, and muscular strength converge. Our research highlights the clinical importance of addressing low bone mass and enhancing muscle strength as integral components of managing spinal disorders in this demographic. The association of GT with coronal malalignment, SVA, and lower LLL, independent of grip strength or BMD, reaffirms its value as a stable metric for evaluating spinal alignment. These insights have significant implications for clinical practice, particularly in tailoring management strategies, realignment brace design, and surgical planning for older women with spinal disorders. Future research should explore these relationships in a broader population and consider additional factors such as dynamic balance and knee joint parameters.

## 5. Conclusions

In summary, our findings highlight the interplay between spinal alignment, bone quality, and muscle strength in older women with mild symptoms. Addressing low bone mass, improving muscle strength, and ensuring proper posture are essential for managing spinal health in this demographic. GT emerges as a reliable metric for assessing spinal alignment, aiding in clinical decision-making, realignment brace design, and surgical planning for this patient group. Our study offers valuable contributions to the understanding of spinal alignment in mild-symptomatic older women. These findings can inform clinical decision-making, improve patient outcomes, and facilitate more effective management of spinal disorders in this population.

## Figures and Tables

**Figure 1 bioengineering-11-00493-f001:**
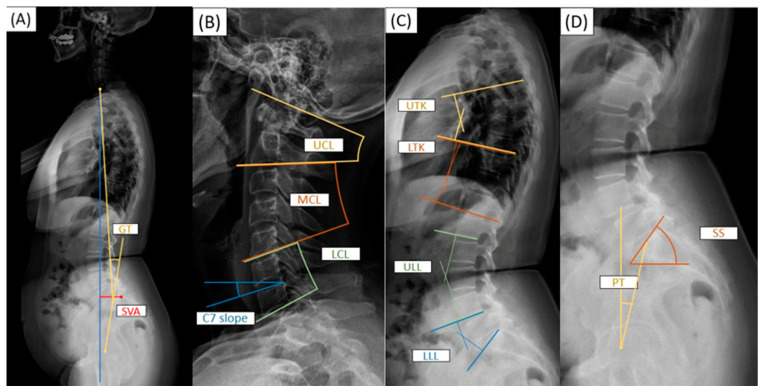
The estimation of spinal sagittal parameters. (**A**) Whole-spine sagittal parameters: GT as global tilt angle and SVA as C7S1 sagittal vertical axis distance; (**B**) cervical spine sagittal parameters: UCL as upper cervical lordotic angle, MCL as middle cervical lordotic angle, LCL as lower cervical lordotic angle, and C7 slope as C7 slope angle; (**C**) thoracolumbar spine sagittal parameters: UTK as upper thoracic kyphotic angle, LTK as lower upper thoracic kyphotic angle, ULL as upper lumbar lordotic angle, and LLL as lower lumbar lordotic angle. (**D**) Spinopelvic sagittal parameters: SS as sacral slope angle and PT as pelvic tilt angle.

**Table 1 bioengineering-11-00493-t001:** Demographics of old females with low bone mass (*n* = 189).

Item	Barthel Index			
	≤80	>80	Total	*p* Value
*n*	55	134	189	
Age	70.63 ± 8.09	68.90 ± 6.91	69.41 ± 7.29	0.140
Postmenopausal period	19.63 ± 7.00	19.56 ± 8.48	19.58 ± 8.06	0.956
BMI	23.82 ± 3.28	24.48 ± 3.74	24.28 ± 3.62	0.255
Grip strength	16.66 ± 4.61	20.01 ± 4.29	19.04 ± 4.63	<0.001 *
Average T score	−2.10 ± 1.00	−1.67 ± 0.65	−1.79 ± 0.79	0.002 *
Coronal malalignment	17 (30.9%)	34 (25.4%)	51 (27.0%)	0.436
Sagittal Parameters				
GT	24.80 ± 9.16	19.25 ± 9.08	20.87 ± 9.42	<0.001 *
SVA	45.99 ± 30.44	33.43 ± 24.46	37.09 ± 26.87	0.003 *
UCL	36.11 ± 6.13	36.56 ± 8.86	36.43 ± 8.14	0.734
MCL	3.48 ± 7.65	4.26 ± 7.20	4.03 ± 7.33	0.509
LCL	6.80 ± 7.06	6.47 ± 7.20	6.57 ± 7.14	0.776
C7 slope	25.61 ± 8.87	24.31 ± 7.71	24.69 ± 8.06	0.319
UTK	−18.43 ± 6.60	−17.99 ± 8.29	−18.12 ± 7.82	0.729
LTK	−5.89 ± 5.69	−6.48 ± 7.16	−6.31 ± 6.75	0.586
ULL	14.35 ± 13.54	16.28 ± 12.54	15.72 ± 12.84	0.349
LLL	28.15 ± 8.00	29.08 ± 9.86	28.81 ± 9.34	0.535
SS	30.61 ± 9.40	33.63 ± 9.88	32.75 ± 9.81	0.055
PT	21.92 ± 7.37	17.36 ± 7.29	18.69 ± 7.59	<0.001 *

The data are presented as *n*, or mean ± standard deviation. * A *p* value < 0.05 was considered statistically significant after the test. BMI = body mass index; GT = global tilt; SVA = sagittal vertical axis; UCL = ulnar collateral ligament; MCL = medial collateral ligament; LCL = lateral collateral ligament; UTK = upper thoracic kyphosis; LTK = lower thoracic kyphosis; ULL = upper lumbar lordosis; LLL = lower lumbar lordosis; SS = sacral slope; PT = pelvic tilt.

**Table 2 bioengineering-11-00493-t002:** Clinical or radiographic factors associated with the Barthel index (*n* = 189).

Item	Crude		Adjusted	
	β (95% CI)	*p* Value	β (95% CI)	*p* Value
Postmenopausal period	−0.09 (−0.21, 0.03)	0.140		
BMI	0.17 (−0.09, 0.44)	0.194		
Grip strength	0.68 (0.50, 0.86)	<0.001 *	0.53 (0.35, 0.71)	<0.001 *
Average T score	2.58 (1.57, 3.60)	<0.001 *	1.49 (0.56, 2.43)	0.002 *
Global tilt angle	−0.25 (−0.35, −0.16)	<0.001 *	−0.16 (−0.25, −0.08)	<0.001 *
Adjusted R^2^ = 0.33				

The data are presented as β (95% CI). * A *p* value < 0.05 was considered statistically significant after the test. BMI = body mass index. Model: simple and multiple linear regression.

**Table 3 bioengineering-11-00493-t003:** Clinical or radiographic factors associated with global tilt angle (*n* = 189).

Item	Crude		Adjusted	
	β (95% CI)	*p* Value	β (95% CI)	*p* Value
Menopause period	0.30 (0.14, 0.46)	<0.001 *	0.07 (−0.07, 0.21)	0.312
BMI	0.05 (−0.32, 0.43)	0.780		
Grip strength	−0.47 (−0.75, −0.18)	0.002 *	−0.22 (−0.46, 0.02)	0.076
Coronal malalignment (Yes vs. No)	5.35 (2.40, 8.31)	<0.001 *	2.33 (0.02, 4.79)	0.048 *
Average T score	−2.29 (−3.79, −0.79)	0.003 *	−0.82 (−2.05, 0.41)	0.190
Sagittal Parameters				
SVA	0.22 (0.18, 0.26)	<0.001 *	0.19 (0.15, 0.24)	<0.001 *
UCL	0.15 (−0.01, 0.32)	0.073		
MCL	0.16 (−0.03, 0.34)	0.093		
LCL	0.12 (−0.07, 0.31)	0.203		
C7 slope	0.28 (0.11, 0.44)	0.001 *	−0.08 (−0.23, 0.07)	0.306
UTK	−0.14 (−0.31, 0.03)	0.111		
LTK	−0.14 (−0.34, 0.07)	0.185		
ULL	0.06 (−0.05, 0.16)	0.288		
LLL	−0.22 (−0.36, −0.08)	0.003 *	−0.11 (−0.22, −0.01)	0.043 *
Adjusted R^2^ = 0.47				

The data are presented as β (95% CI). * A *p* value < 0.05 was considered statistically significant after the test. BMI = body mass index; SVA = sagittal vertical axis; UCL = ulnar collateral ligament; MCL = medial collateral ligament; LCL = lateral collateral ligament; UTK = upper thoracic kyphosis; LTK = lower thoracic kyphosis; ULL = upper lumbar lordosis; LLL = lower lumbar lordosis. Model: simple and multiple linear regression.

## Data Availability

Data are contained within the article.

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
