# Peer review of "Research of Global Tilt and Functional Independence: Insights into Spinal Health of Older Women"

_bioengineering, 2024, doi:10.3390/bioengineering11050493_

Round 1
Reviewer 1 Report
Comments and Suggestions for Authors
Many Methodological Biases exist
(The Authors must see my remarks)

Author Response
This study included 189 elderly women with chronic mild neck and low back pain, and elucidated the relationship between spinal alignment, bone mineral density (BMD), and muscle strength in elderly women with mild neck and low back pain. Research has shown that addressing bone mass, augmenting muscle strength, and ensuring proper posture are critical for spinal health management in this population, with GT serving as a crucial parameter for clinical assessment and intervention planning. The following points should be clarified or modified by the author in the article.
- About word modification
Ans: Thank you for your suggestions. We have revised all our “sex” into “gender”. We also have revised our “correlation” into “association”.
- About the lack of reference
Ans: Thank you for your suggestion. We have added the adequate references to the sentences you pointed out.
- About sagittal parameters estimation with the reference.
Ans: Thank you for your suggestions. We have added the estimation methods of every parameter into the third paragraph of Material and Methods section:” GT is determined by the angle between C7, the center of the sacrum and the line be-tween the center of the femoral heads and the center of the sacrum. SVA is the length of a horizontal line connecting the posterior superior sacral end plate to a vertical plumbline dropped from the centroid of the C7 vertebral body. The cervical lordotic angle was divided into UCL,MCL and LCL. UCL is the angle, measured in degrees, formed between the cranial line (C0) and the lower endplate of C2. MCL is the angle, measured in degrees, formed between the lower endplate of C2 and the lower endplate of C5. LCL is the angle, measured in degrees, formed between the lower endplate of C5 and the lower endplate of C7. C7 slope is determined by measuring the angle between a horizontal reference line and a line that runs parallel to the upper endplate of C7. The thoracic kyphotic angle was divided into UTK and LTK. UTK is the angle, measured in degrees, formed between the lower endplate of T5 and the lower endplate of T9. LTK is the angle, measured in degrees, formed between the lower endplate of T9 and the lower endplate of T12. The lumbar lordotic angle was divided into ULL and LLL. ULL refers to the angle, measured in degrees, formed between the lower endplate of L1 and the lower endplate of L4. LLL is the angle, measured in degrees, formed between the lower endplate of L4 and the upper endplate of the S1. PI is the angle between the line perpendicular to the sacral endplate at its midpoint and a line connecting this point to the axis of the femoral head. SS is the horizontal and sacral plate angle. PT is the angle between a vertical reference line and a line from the midpoint of the sacral endplate to the femoral rotational axis.” We have added the references related to the estimation of the sagittal alignment parameters. We also add a figure for the description of the measurement of the sagittal parameters.
- Weak and unreliable model... A Logistic Regression analysis should be more useful. ''t'' test(s)? If so, were the distributions normal? The model of table 2 and table 3.
Ans: Thank you for your reminding. Because most of our parameters are continuous variables, we use simple and multiple linear regression instead of logistic regression analysis and presented their results as table 2 and table 3. ). Chi-square tests and independent t tests / Wilcoxon rank-sum test were used to evaluate the associations between demographic characteristics and the Bathel index score and the results were shown as table 1. The statistical method section has been modified as follows:” All statistical analyses were conducted using IBM SPSS Statistics for Windows (version 23.0; IBM, Armonk, NY, USA). Chi-square tests and independent t tests / Wilcoxon rank-sum test (depending on whether the normality assumption holds or not) were used to evaluate the associations between demographic characteristics and the Bathel index score. Simple and multiple linear regression analysis was conducted to examine the associations between the Barthel index and grip strength, average T score, and GT along with other clinical parameters. The same analysis was conducted to investigate the associations between GT and both average T score and grip strength as well as other clinical and sagittal parameters. All p values were obtained using two-sided tests, with p < 0.05 indicating statistical significance.” In addition, the model of table 2 and table 3 has been added to the footnote of each table as a simple and multiple linear regression.
- How did the Authors determine the study size? Protocol? References? Inclusion / Exclusion Criteria?
Ans: Thanks for your reminding. The study size: According to Bonett et al, the recommended sample size should be at least 50 + 8* k. K represents for the number of predictors considered in the multiple linear models. For Table 2 (Barthel Index), the minimum sample size should be 50+8*5=90. For Table 3 (Global tilt angle), the minimum sample size should be 50+8*14=162. Thus, our sample size (n=189) fulfilled the minimum requirement for sample size calculation [20].
- BONETT, D. G., & WRIGHT, T. A. (2011). Sample size requirements for multiple regression interval estimation. Journal of Organizational Behavior, 32(6), 822–830. http://www.jstor.org/stable/41415703
The protocol: After providing written informed consent, women aged 65 or older who were seeking consultations for bone health between August 2019 and July 2023 were recruited for bone health evaluation from the Orthopedic Department of our Hospital. These patients underwent dual-energy X-ray absorptiometry for bone mineral density (BMD) measurements. Upon agreeing to participate in the study, the patients provided information on their age, menopausal status, and BMI. We then arranged full-length standing antero-posterior and lateral radiography examination for them for further measurement of the radiographic parameters, measured their grip strength, which was determined as the mean of triplicate measures of bilateral hand strength, and evaluated their Barthel index score, which was a standardized assessment tool that quantifies an individual’s functional independence by scoring their ability to perform essential activities of daily living.
Inclusion/Exclusion criteria: The inclusion criteria were as follows: 1. Female gender older than 65 years; 2. Having the intermittent minimal-mild neck or back pain , with a pain severity measured between 0 and 3 on the Visual Analog Scale (VAS); 3. Having a normal walking status; and 4. Having low bone mass, defined by a T score of below −1.0 at specific locations, including the lumbar spine (average), right hip (femoral neck or total femur), and left hip (femoral neck or total femur). The exclusion criteria were as follows: 1. Those who previously underwent spine surgery, femoral neck fracture, artificial hip or knee replacement, or experienced acute spinal trauma within the previous 6 months; 2. Those who had a history of stroke or had a history of a neurological disorder such as Parkinson’s or Alzheimer’s disease; 3. Those who cannot tolerate or agree to receiving the whole spine standing lateral view X-ray examination.
- The demographic data in Table 1 are suggested to be grouped into high Barthel index and low Barthel index.
Ans: Thanks for your suggestion. We have revised our table 1 and modified the first subsection of the Result section as below:” There were 189 elderly females with low bone mass, segmented by their Barthel Index scores into two groups: those with scores of 80 or below and those with scores above 80 (Table 1). This distinction helps in understanding the relationship between various clinical parameters and functional independence as measured by the Barthel Index. Of the total participants, 55 had a Barthel Index of 80 or less, and 134 scored above 80. The overall average age was 69.41 years, with the lower Barthel group being slightly older (70.63 years) compared to the higher score group (68.90 years), although this age dif-ference was not statistically significant (p=0.140)(Table 1). Both groups had a similar duration of post-menopausal period, averaging around 19.58 years across the cohort, with no significant difference (p=0.956). There was a slightly higher average BMI in the group with a higher Barthel score (24.48) compared to those with a lower score (23.82), but this difference was not statistically significant (p=0.255). Grip strength was signifi-cantly higher in the group with Barthel scores over 80 (20.01) compared to those with scores of 80 or less (16.66), indicating an association between higher physical strength and better functional performance (p<0.001)(Table 1). This measure of bone density was significantly higher (indicating better bone quality) in the higher Barthel score group (-1.67) compared to the lower score group (-2.10), with a p-value of 0.002. Both GT and SVA showed significant differences between the groups (Table 1). Higher GT (24.80) and SVA (45.99) were observed in participants with lower Barthel scores, indicating more pronounced spinal misalignment associated with decreased functional independence (p<0.001 for GT and p=0.003 for SVA). Similarly, PT was significantly higher in the lower Barthel group (21.92), further illustrating the relationship between pelvic alignment and functional capacity (p<0.001) (Table 1) .”
- No subsection in Discussion section
Ans: Thank you for your suggestion. We have removed the subsection from Discussion section and reorganize the Discussion.
- Avoid repetition: “To our knowledge, this is the first study to investigate the correlations between spinopelvic parameters and BMI, BMD, and grip strength. To address a major gap in the literatures, we focused on the major patient group in this aged society as mild-symptomatic older women with mild neck and back pain. Our findings revealed that within this cohort of old females experiencing mild symptoms, the Barthel index correlated with BMD, GT, and grip strength and that GT correlated with coronal malalignment, SVA, and LLL. Overall, these findings provide valuable insights into improving the function of and offering advice for this particular population.”
Ans: Thanks for your reminding. We have revised this paragraph as follows:” To our knowledge, this is the first study to investigate the associations between spinopelvic parameters and BMI, BMD, and grip strength. To address the gap in the literature, we focused on the major patient group in this aged society as mild-symptomatic older females. Our findings revealed that the Barthel index associated with BMD, GT, and grip strength and that GT associated with coronal malalignment, SVA, and LLL within this cohort. Overall, these findings provide valuable in-sights into improving the function of and offering advice for this particular population.”
- Limitation: Methodological Biases exist.
Ans: Thanks for your reminding. We have added the limitation as follows:” First, the methodological bias exist in this study . Although our study's inclusion criteria focused on a specific age, gender, and demographic group, this approach was strategic. By targeting precisely defined subgroups, we aimed to mitigate the effects of variations in these characteristics, thereby enhancing the robustness and relevance of our findings to the target demographic. However, this specificity limits the generalizability of our results to broader populations. Although we employed standardized tools such as the Barthel Index and DEXA scans for assessing functional independence and bone density, respectively, there might still be inconsistencies in how these measurements were applied or interpreted across different settings or examiners. While we adjusted for several potential confounders in our analyses, there may be additional unmeasured variables that could influence the outcomes, such as socioeconomic status, dietary factors, or previous medical interventions.”
- Reduce this paragraph: “In this study, we investigated the interplay between spinal alignment parameters and functional indices in older women with mild neck and back pain. Our key findings reveal a significant correlation between the functional parameter, the Barthel index and GT, BMD, and grip strength. This underlines the multifaceted nature of spinal health, where structural alignment, bone quality, and muscular strength converge. Our research highlights the clinical importance of addressing low bone mass and enhancing muscle strength as integral components of managing spinal disorders in this demographic. Encouraging appropriate posture, particularly in sitting positions, emerges as a crucial strategy to prevent malalignment. The correlation of GT with coronal malalignment, SVA, and lower LLL, independent of grip strength or BMD, reaffirms its value as a stable metric for evaluating spinal alignment. These insights have significant implications for clinical practice, particularly in tailoring management strategies, re-alignment brace design and surgical planning for older women with spinal disorders. By elucidating the complex relationships between spinal alignment parameters and functional capabilities, our study contributes to a more comprehensive understanding of spinal health in older women. Future research should explore these relationships in a broader population and consider additional factors such as dynamic balance and knee joint parameters. Such studies could provide a more holistic view of spinal health and its management in diverse patient groups.”
Ans: Thank you for your suggestion. We have modified this paragraph as follows:” Despite of these limitations, our study still revealed a significant association between the functional parameter, the Barthel index and GT, BMD, and grip strength. This underlines the multifaceted nature of spinal health, where structural alignment, bone quality, and muscular strength converge. Our research highlights the clinical im-portance of addressing low bone mass and enhancing muscle strength as integral components of managing spinal disorders in this demographic. The association of GT with coronal malalignment, SVA, and lower LLL, independent of grip strength or BMD, reaffirms its value as a stable metric for evaluating spinal alignment. These insights have significant implications for clinical practice, particularly in tailoring management strategies, re-alignment brace design and surgical planning for older women with spinal disorders. Future research should explore these relationships in a broader population and consider additional factors such as dynamic balance and knee joint parameters.“

Reviewer 2 Report
Comments and Suggestions for Authors
This study included 189 elderly women with chronic mild neck and low back pain, and elucidated the relationship between spinal alignment, bone mineral density (BMD), and muscle strength in elderly women with mild neck and low back pain. Research has shown that
addressing bone mass, augmenting muscle strength, and ensuring proper posture are critical for spinal health management in this population, with GT serving as a crucial parameter for clinical assessment and intervention planning. The following points should be clarified or modified by the author in the article.
1. Why use the Barthel index instead of other similar scores.
2. The demographic data in Table 1 are suggested to be grouped into high Barthel index and low Barthel index.
3. The paper should add the measurement method of Sagittal Parameters and the measurement diagram of Sagittal Parameters of the spine.
4. Sagittal Parameters are measured by multiple people or single people? If measured by multiple people, the reliability of measurements between observers should be shown in a table.
Author Response
Comments and Suggestions for Authors (Reviewer 2)
This study included 189 elderly women with chronic mild neck and low back pain, and elucidated the relationship between spinal alignment, bone mineral density (BMD), and muscle strength in elderly women with mild neck and low back pain. Research has shown that addressing bone mass, augmenting muscle strength, and ensuring proper posture are critical for spinal health management in this population, with GT serving as a crucial parameter for clinical assessment and intervention planning. The following points should be clarified or modified by the author in the article.
- Why use the Barthel index instead of other similar scores.
Ans: Thank you for your question regarding our choice of the Barthel Index. We selected it due to its specific relevance to elderly populations, assessing crucial daily activities pertinent to our study's demographic. Its sensitivity for mild to moderate disability aligns well with our cohort of elderly women with mild symptoms. The Index's ease of administration and widespread use facilitate comprehensive, efficient data collection and comparative analysis. Furthermore, its clinical relevance supports practical applications in managing spinal health, making it an ideal tool for evaluating functional independence in our research.
- The demographic data in Table 1 are suggested to be grouped into high Barthel index and low Barthel index.
Ans: Thank you for your suggestion. We have made the modification of table 1 and modified the description of it in Result section.
- The paper should add the measurement method of Sagittal Parameters and the measurement diagram of Sagittal Parameters of the spine.
Ans: Thank you for your reminding. We have added the following sentences into the third paragraph of Material and Methods section:” GT is determined by the angle between C7, the center of the sacrum and the line be-tween the center of the femoral heads and the center of the sacrum. SVA is the length of a horizontal line connecting the posterior superior sacral end plate to a vertical plumbline dropped from the centroid of the C7 vertebral body. The cervical lordotic angle was divided into UCL,MCL and LCL. UCL is the angle, measured in degrees, formed between the cranial line (C0) and the lower endplate of C2. MCL is the angle, measured in degrees, formed between the lower endplate of C2 and the lower endplate of C5. LCL is the angle, measured in degrees, formed between the lower endplate of C5 and the lower endplate of C7. C7 slope is determined by measuring the angle between a horizontal reference line and a line that runs parallel to the upper endplate of C7. The thoracic kyphotic angle was divided into UTK and LTK. UTK is the angle, measured in degrees, formed between the lower endplate of T5 and the lower endplate of T9. LTK is the angle, measured in degrees, formed between the lower endplate of T9 and the lower endplate of T12. The lumbar lordotic angle was divided into ULL and LLL. ULL refers to the angle, measured in degrees, formed between the lower endplate of L1 and the lower endplate of L4. LLL is the angle, measured in degrees, formed between the lower endplate of L4 and the upper endplate of the S1. PI is the angle between the line perpendicular to the sacral endplate at its midpoint and a line connecting this point to the axis of the femoral head. SS is the horizontal and sacral plate angle. PT is the angle between a vertical reference line and a line from the midpoint of the sacral endplate to the femoral rotational axis. Four physicians independently measured the parameters from the plain films. The intraclass correlation coefficient of the measurements of them was 0.88 and 0.81 for intraobserver and interobserver agreement, respectively.” We also add a figure for the description of the measurement of the sagittal parameters.
- Sagittal Parameters are measured by multiple people or single people? If measured by multiple people, the reliability of measurements between observers should be shown in a table.
Ans: Thank you for your reminding. We will present this information by the descriptive sentences. We have added the following sentences into the end of the third paragraph of Material and Methods section:” Four physicians independently measured the parameters from the plain films. The intraclass correlation coefficient of the measurements of them was 0.88 and 0.81 for intraobserver and interobserver agreement, respectively.”
